# PLVAP as an Early Marker of Glomerular Endothelial Damage in Mice with Diabetic Kidney Disease

**DOI:** 10.3390/ijms24021094

**Published:** 2023-01-06

**Authors:** Elena E. Wolf, Anne Steglich, Friederike Kessel, Hannah Kröger, Jan Sradnick, Simone Reichelt-Wurm, Kathrin Eidenschink, Miriam C. Banas, Eckhard Wolf, Rüdiger Wanke, Florian Gembardt, Vladimir T. Todorov

**Affiliations:** 1Experimental Nephrology, Division of Nephrology, Department of Internal Medicine III, University Hospital Carl Gustav Carus, TU Dresden, 01307 Dresden, Germany; 2Department of Nephrology, University Hospital Regensburg, 93053 Regensburg, Germany; 3Molecular Animal Breeding and Biotechnology, Gene Center, Ludwig-Maximilians-Universität München, 81377 Munich, Germany; 4German Center for Diabetes Research (DZD), 85764 Neuherberg, Germany; 5Institute of Veterinary Pathology, Center for Clinical Veterinary Medicine, Ludwig-Maximilians-Universität München, 80539 Munich, Germany

**Keywords:** diabetic kidney disease, endothelial damage, PLVAP, diabetic models, glomerular hypertrophy

## Abstract

Plasmalemma vesicle-associated protein (PLVAP) is the main component of endothelial diaphragms in fenestrae, caveolae, and transendothelial channels. PLVAP is expressed in the adult kidney glomerulus upon injury. Glomerular endothelial injury is associated with progressive loss of kidney function in diabetic kidney disease (DKD). This study aimed to investigate whether PLVAP could serve as a marker for glomerular endothelial damage in DKD. Glomerular PLVAP expression was analyzed in different mouse models of DKD and their respective healthy control animals using automatic digital quantification of histological whole kidney sections. Transgenic mice expressing a dominant-negative GIP receptor (GIPR^dn^) in pancreatic beta-cells as a model for diabetes mellitus (DM) type 1 and black and tan brachyuric (BTBR) *ob/ob* mice, as a model for DM type 2, were used. Distinct PLVAP induction was observed in all diabetic models studied. Traces of glomerular PLVAP expression could be identified in the healthy control kidneys using automated quantification. Stainings for other endothelial injury markers such as CD31 or the erythroblast transformation-specific related gene (ERG) displayed no differences between diabetic and healthy groups at the time points when PLVAP was induced. The same was also true for the mesangial cells marker α8Integrin, while the podocyte marker nephrin appeared to be diminished only in BTBR *ob/ob* mice. Glomerular hypertrophy, which is one of the initial morphological signs of diabetic kidney damage, was observed in both diabetic models. These findings suggest that PLVAP is an early marker of glomerular endothelial injury in diabetes-induced kidney damage in mice.

## 1. Introduction

The capillary endothelium is characterized by a wide structural and functional heterogeneity to adapt to specific metabolic needs in various organs. Fenestrae, caveolae, and transendothelial channels are part of the capillary endothelium. These structures allow for the transendothelial transport of water and selected molecules between the capillary blood and interstitial fluid [1,2,3,4]. The fenestrae, caveolae, and transendothelial channels are all featured by diaphragms at their surface area. Plasmalemma vesicle-associated protein (PLVAP) is the only structural component that has been identified as part of the diaphragm. Therefore, it has a critical role to play in maintaining the integrity of microvascular permeability [5,6]. PLVAP is a type II integral membrane glycoprotein that is specifically expressed in endothelial cells of the blood and lymphatic vessels [5,7,8,9]. In the adult kidney, PLVAP is detectable in the fenestrated endothelium of peritubular capillaries and the ascending vasa recta. In glomerular capillaries, PLVAP is absent because the glomerular endothelium lacks diaphragms in their fenestrae [5]. However, glomerular expression of PLVAP has been reported in models of renal endothelial injury such as transplant glomerulopathy, Thy-1.1 glomerulonephritis, and thrombotic microangiopathy in mice with defective Gsα/cAMP signaling in renin cells [10,11,12].

Diabetes mellitus (DM) is the leading cause of chronic kidney disease and end-stage renal disease worldwide [13,14]. In DM, hyperglycemia activates a complex cascade network of signal molecules, cytokines, and growth factors that eventually leads to the formation of reactive oxygen species and advanced glycosylated end products. In the kidney, these pathophysiological processes induce diabetic kidney disease (DKD), which is characterized by multiple structural changes leading to progressive loss of kidney function [14,15]. Initial histopathological alterations in glomeruli include endothelial damage, thickening of the glomerular basement membrane, extracellular matrix accumulation, mesangial expansion, and loss of podocytes, while in later stages segmental or global sclerosis is dominant [14].

From a functional point of view, the early stages of DKD are featured by glomerular hyperfiltration and hypertrophy, while in advanced stages the glomerular filtration rate invariably declines [14,16,17,18].

Characteristic alterations in the glomerular endothelium in early DKD are endothelial cell dysfunction and the disturbance of the surface glycocalyx. However, these parameters are hardly applicable as markers for diagnostic use because indirect readouts and sophisticated methodology are used for their evaluation. On the other hand, the hyperglycemia-induced deterioration of endothelial cell function is causally related to the developing failure of the filtration barrier and progressive albuminuria, which pave the way for the development of kidney failure in DKD [19,20,21]. Therefore, we aimed to identify an early morphological marker for glomerular endothelial damage in DKD. In the present study, we hypothesized that PLVAP may serve as a marker for glomerular endothelial injury. To this end, we investigated the glomerular PLVAP expression of various mouse models with DKD using the automated evaluation of immunostained kidney slices. We found that induced glomerular PLVAP production is detectable in diabetic kidney tissue before other endothelial injury markers and around the onset of other early signs of glomerular damage such as hypertrophy or matrix accumulation.

## 2. Results

### 2.1. Glomerular PLVAP Expression Is Induced in DM Type 1 and Type 2 Mouse Models

We obtained kidneys from different transgenic mouse models of DM: mice expressing a dominant-negative GIP receptor (GIPR^dn^) which served as insulin-deficient DM type 1 model and black and tan brachyuric (BTBR) *ob/ob* mice, which represent a mouse model for DM type 2. The time points of kidney tissue harvesting were chosen to correspond to the early stages of DKD [22,23,24,25,26]. A distinct PLVAP signal was detectable in the glomeruli of GIPR^dn^ transgenic mice (Figure 1). Induced glomerular PLVAP protein expression could be confirmed in an independent insulin-deficient non-transgenic DM type 1 model (STZ-treated wildtype mice, Appendix A). PLVAP was also significantly upregulated in glomeruli of BTBR *ob/ob* mice (Figure 1). Throughout these experiments, PLVAP appeared to always be expressed in the glomerular endothelium (Appendix A). The fluorescent labelling of the PLVAP protein revealed low but consistent glomerular expression in all healthy control animals (Figure 1).

### 2.2. Glomerular Expression of Endothelial CD31 and Erythroblast Transformation-Specific Related Gene (ERG) Is Not Decreased in DM Type 1 and Type 2 Mouse Models

We analyzed whether the glomerular expression of CD31 and ERG, which are dysregulated upon endothelial injury, is affected at the time points when PLVAP was already induced in our transgenic mouse models of DM. Neither CD31 (Figure 2) nor ERG expression (Figure 3) was different between the glomeruli of healthy and diabetic animals.

### 2.3. Glomerular Expression of Mesangial and Podocyte Markers in DM Type 1 and Type 2 Mouse Models

To detect possible signs of injury in the glomerular mesangial cells and podocytes, which are also affected in DKD, we assessed the expression of the mesangial cell marker α8Integrin and the podocyte marker nephrin. α8Integrin expression was not different between the glomeruli of healthy and diabetic mice (Figure 4). However, glomerular matrix accumulation, which is largely dependent on mesangial cells, was enhanced under diabetic conditions (Appendix A). The podocyte cell marker nephrin was decreased in the glomeruli of BTBR *ob/ob* mice (Figure 5). Accordingly, proteinuria as an indirect marker of podocyte dysfunction was detected in both transgenic diabetes models at the observation time point (Appendix A).

### 2.4. General Glomerular Morphology in DM Type 1 and Type 2 Mouse Models

We also evaluated the glomerular morphology (glomerular cross-section area and cell density) in the studied mouse models of DKD. In the GIPR^dn^ mouse model, the glomerular area was increased and the cell density tended to decrease (*p* = 0.15, Figure 6). The glomerular area was also increased, while the cell density was significantly diminished in the BTBR *ob/ob* DM type 2 mouse model (Figure 6).

## 3. Discussion

In the present study, we provide evidence that PLVAP is an early marker of glomerular endothelial damage in the kidneys of mice with DM. DM is a chronic disease with has undergone pandemic expansion. Since 1980, the number of diabetic patients rose more than fourfold to over 422 million worldwide [27]. DM is a metabolic disorder disturbing virtually all body organs and systems. The kidneys are particularly affected by DM. About 25–40% of DM patients develop DKD over time [14,28]. Therefore, DKD is both the most common kidney disease and the most common cause of end-stage renal disease worldwide [13,29]. A general problem in the diagnosis of DKD is the limited number of markers for early renal morphological changes. Within the kidney, the endothelium is considered to be the first structural compartment that is progressively affected by hyperglycemia in DM. In the context of the diabetic kidney, alterations in the glomerular microendothelium are particularly indicative for at least two reasons. Firstly, the endothelial cells in the glomerulus are subjected not only to hyperglycemic, but also to increased flow and pressure stress due to glomerular hyperfiltration under diabetic conditions. This dual distress predestines the glomerular endothelium of the diabetic kidney for early injury. Secondly, the progression of glomerular endothelial damage parallels the development of albuminuria, fibrosis, and irreversible loss of kidney function in DKD [30,31,32,33]. Therefore, the identification of morphological markers of early endothelial diabetic injury in the glomerulus is of particular translational importance. One of the earliest signs of microvascular damage in DM is the deterioration of the endothelial surface layer (ESL) [21,30,34]. The ESL is composed primarily of the glycocalyx, which consists of cell membrane-bound proteoglycans and glycoproteins [30,35]. Thinning and loss of ESL are typical findings across the microendothelium in diabetes patients as well as in diabetic animal models. However, the glycocalyx is destroyed during standard tissue fixation for histological staining. Electron microscopy and indirect methods for assessment of the ESL are either sophisticated or have limited diagnostic potential.

We here report on PLVAP as an early glomerular-specific histological marker of microendothelial damage in different diabetic mouse models. PLVAP is expressed in the glomerular endothelium after injury [10,11,12]. In contrast to some reports [5,10,11], we found traces of PLVAP protein in the glomeruli of healthy mice by using an unbiased automatic quantification of whole histological kidney sections. Although the fluorescent signal for glomerular PLVAP in control animals appeared weak, it was definitely above the background threshold because it was controlled by a staining without a primary antibody (negative control, see Materials and Methods). Moreover, others and us have already observed glomerular PLVAP in adult healthy mice [12,36]. Therefore, we are convinced that PLVAP is expressed in the glomeruli of mature healthy mice, albeit at a very low level. One possible explanation for this finding might be that PLVAP marks areas of subtle and probably reversible damage developing in the glomerular endothelium during adult life. The strong expression of PLVAP was observed in the glomeruli of transgenic mouse strains representing well-described and generally accepted models for DM type 1 and type 2 [22,23,24,25,26]. These results fit the current paradigm postulating that PLVAP production is induced in diseased glomeruli [10,11,12]. Our findings are also in line with an earlier study reporting the induction of glomerular PLVAP expression in diabetic BMP and Activin Membrane-Bound Inhibitor (BAMBI)-deficient mice [37]. PLVAP was also detected at a later time point in the glomeruli of BTBR *ob/ob* mice [38]. A very recent paper impressively demonstrated induced PLVAP expression in glomerular capillaries of BTBR *ob/ob* mice and diabetic patients, corroborating the translational significance of PLVAP as a diagnostic marker in DKD [39]. However, it is still not clear whether PLVAP is mechanistically involved in the processes of injury or regeneration upon initial damage. Currently, arguments for both assumptions exist. Thus, on the one hand, PLVAP could facilitate the infiltration of immune cells to support tissue damage [1,5,40]. On the other hand, PLVAP is expressed in the glomerular endothelium during nephrogenesis [11], and embryonic processes operating during fetal life could generally be re-activated to repair damaged organs in adulthood. Moreover, it is known that endothelial fenestration in glomeruli is diminished early in DKD [19,39]. Since PLVAP is necessary for the functional and structural integrity of fenestrae [5,40,41,42,43,44], one may speculate that its induced expression counteracts the fenestral disintegration in the glomerular endothelium during DM.

Although PLVAP induction is not specific for diabetic glomerular damage, it might prove very important as an early marker of endothelial and glomerular injury in DKD. At the time points studied, hyperglycemia already persisted in the diabetic mouse strains for 8 to 20 weeks [22,23,24,25]. At these time points, PLVAP was induced before alterations in other endothelial-specific injury markers such as CD31 or ERG. We also observed an increased glomerular cross-section area in the diabetic mouse models, which reflects glomerular hypertrophy as a typical early sign of DKD. However, we detected glomerular PLVAP at time points when some specific morphological markers of glomerular injury could not be uniformly confirmed by (immuno)histochemical stainings. Thus, the mesangial cell marker α8Integrin was not diminished despite increased PAS positivity, suggesting an initial mesangial involvement. Using automated quantification, we found that the podocyte marker nephrin was decreased only in the glomeruli of BTBR *ob/ob* mice, while proteinuria was present in both diabetic models. Similar to the findings in the mesangium, this constellation infers a beginning podocyte dysfunction.

Our study is limited by the use of a sole immunohistochemistry-based evaluation method and a single time point for analysis of the different mouse models. However, it consistently demonstrates that PLVAP could be used as a histological marker of glomerular endothelial injury in DM that seems superior to other endothelial markers such as CD31 or ERG. Interestingly, circulating PLVAP was reported as a serum marker of celiac disease-associated liver injury in humans [45]. This finding opens the possibility for using PLVAP levels in body fluids like serum or urine as an unsophisticated diagnostic readout of microvascular injury in diabetic patients. Such an intriguing perspective could be addressed by future studies.

In summary, we identified PLVAP as a glomerular endothelial marker induced early in different mouse models of DKD. Since PLVAP is a general marker of glomerular endothelial injury and because of the high prevalence of DM, the histological evaluation of the glomerular PLVAP expression could become an important tool for the morphological diagnosis of early DKD.

## 4. Materials and Methods

### 4.1. Animals

All animal experiments were consistent with the German Animal Welfare Act and were approved by the local authorities. Animal experiments were performed in accordance with the National Institutes of Health (NIH) *Guide for the Care and Use of Laboratory Animals* (NIH Pub. No. 85-23, Revised 2011). Mice were housed in standard cages with a 12/12 light/dark cycle and had ad libitum access to standard food and water.

### 4.2. GIPR^dn^ Mice

Twenty-four-week-old CD1 mice expressing a dominant-negative glucose-dependent insulinotropic peptide (GIP)-receptor (GIPR^dn^) in pancreatic beta cells were used as an insulin-deficient transgenic model of DM type 1, as described in detail elsewhere [23,24]. The mice develop a disturbance of the pancreatic beta-cells, with an early reduction in insulin secretion. This mouse line exhibits hyperglycemia, proteinuria, and some typical morphological changes of DKD at later time points (Appendix A) [23,24].

### 4.3. BTBR ob/ob Mice

Paraffin-embedded kidney sections from 16-week-old black and tan brachyuric (BTBR) *ob/ob* mice and their controls were obtained from the University Hospital in Regensburg. These mice have a mutation in the leptin gene and serve as a model of DM type 2 with renal involvement [25]. Breeding conditions and organ harvesting were previously described [26]. Due to the deficiency of leptin [22,25], these mice develop insulin resistance with obesity, hyperglycemia, and proteinuria (Appendix A).

### 4.4. High-Dose Streptozotocin-Induced DM in Mice

To induce an additional model of insulin-deficient type 1 diabetes, one high-dose of streptozotocin (STZ, 180 mg/kg in 50 mM sodium-citrate buffer, pH 4.5) was administered by intraperitoneal injection in 6–8 week old C57BL/6J mice. After injection, the mice received water with 10% sucrose. This is a slightly modified protocol recommended by The Diabetic Complications Consortium (www.diacomp.org; accessed on 21 December 2022). We marginally increased the STZ dose (180 mg/kg BW instead of 150 mg/kg BW as recommended) to avoid prolonged fasting intervals. The control mice received the same volume of saline by intraperitoneal injection. Mice were sacrificed and the kidneys were harvested 16–18 weeks after intraperitoneal injection.

### 4.5. Collection and Analysis of Urine and Serum Samples

Urine and serum collection was performed as previously described [12]. Urinary albumin was detected with a mouse albumin ELISA kit (Bethyl Laboratories Inc., Montgomery, TX, USA). For measurement of total urinary protein, the urine was diluted 1:10 and the bicinchoninic acid (BCA) assay was applied, using standard protocols. For colorimetric quantification, the absorption at a wavelength of 562 nm was measured using a Tecan Infinite 200 Pro reader (Tecan Deutschland GmbH, Craisheim, Germany). Blood glucose levels were measured using blood sensor glucometers.

### 4.6. Tissue Sample Preparation

The mice were sacrificed by exsanguination under inhalation anesthesia (2–3% isoflurane (Baxter Deutschland GmbH, Heidelberg, Germany)). The kidneys were harvested after perfusion with 0.9% NaCl to minimize artifacts by erythrocyte autofluorescence in the histological analysis. Th kidneys were cut in half and fixed overnight in 4% paraformaldehyde (PFA) or zinc-fixative (0.05% calcium acetate, 0.5% zinc acetate, 0.5% zinc chloride *w*/*v* in 0.1 M TRIS buffer, pH 7.4) at 4 °C. All samples were embedded in paraffin and cut into 2.5 µm thick sections.

### 4.7. Immunostaining of Histological Sections

For immunofluorescent staining, all samples were permeabilized for 20 min with 0.5% Triton X-100 (Sigma-Aldrich Chemie GmbH, Darmstadt, Germany) and blocked with 5% normal host serum for 30 min. The slides were incubated overnight at 4 °C with diluted primary antibodies, followed by incubation with fluorescence-labeled secondary antibodies for 2 h at room temperature. Further information about the antibodies is provided in Appendix A. Nuclei were stained for 2 min using 1:5000 4′,6-diamidino-2-phenylindole (DAPI; A1001; AppliChem GmbH, Darmstadt, Germany). Periodic acid-Schiff (PAS) staining was performed according to standard protocols on paraffin-embedded whole kidney sections.

### 4.8. Microscopy and Analysis of Histological Sections

The scan of the whole kidney sections was performed by a Zeiss Axio Z1 Slide-Scanner. The automatic analysis of histological sections (“AQUISTO”) was previously established by our laboratory and has been described elsewhere [46]. The algorithm for the automatic image section and data analysis was designed by FIJI software (Version 1.52; [47]), R (Version 3.4.3; [48]) running under RStudio (Version 1.1.463; [49]). For the analysis of glomerular morphology, the algorithm detected glomeruli and their glomerular area in all renal histological sections. Quantification of cells was evaluated based on the DAPI-stained nuclei by automatic analysis with the FIJI-plugin “marker-controlled watershed” from MorphoLibJ (Version 1.4.0; [50]). Cells were classified for marker-positivity in additional channels. For quantification of PLVAP, CD31, nephrin, α8Integrin, and ICAM2, the glomerular marker-positive area was analyzed. Analysis parameters for the automatic evaluation of histological sections are shown in Appendix A. The single values and the standard error of mean (SEM) data are also provided as a Appendix A. Negative controls (without primary antibody) were used throughout all stainings and evaluations. Editing of the scale bar was performed by FIJI [51]. The quantification of glomerular PAS-positive material was carried out either by HistoQuest software (Version 3.0; TissueGnostics, Vienna, Austria) referring to the whole area of the respective glomerulus, or by semiquantitative PAS positivity score [22]. The scoring system ranges from 0 to 4, including healthy glomeruli (score = 0), glomeruli with <25% area with increased PAS positivity (score = 1), glomeruli with 25–50% area with increased PAS positivity (score = 2), glomeruli with 50–75% area with increased PAS positivity (score = 3), and glomeruli with >75% area with increased PAS positivity (score = 4).

### 4.9. Statistical Analysis

The statistical analysis was performed with the non-parametric two-tailed Mann-Whitney U-Test (Graph Pad 6, Graph Pad Software Inc., San Diego, CA, USA). The data are presented as scatter plots with means and SEM. A *p*-value < 0.05 was considered significant.

## Figures and Tables

**Figure 1 ijms-24-01094-f001:**
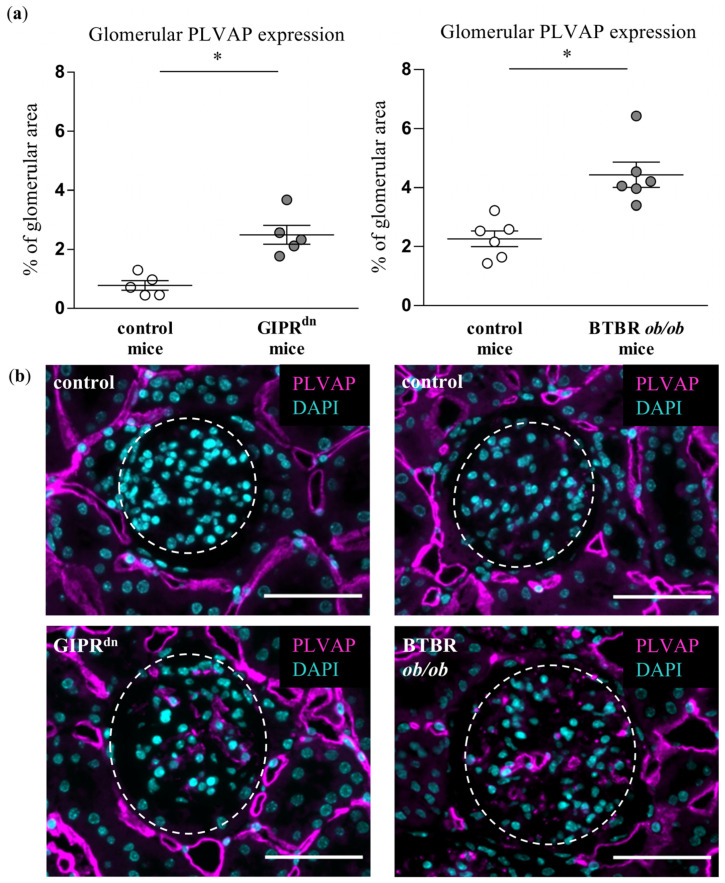
Analysis of plasmalemma vesicle-associated protein (PLVAP) in diabetic kidney disease models. (**a**) Automatic histological evaluation of PLVAP-positive area in glomeruli of GIPR^dn^ mice (**left**, diabetes mellitus type 1 model) and BTBR *ob/ob* mice (**right**, diabetes mellitus type 2 model) compared to their corresponding healthy control mice. Scatter dot plot with means and SEM, * *p* < 0.05, *n* = 5–6. (**b**) Representative images of histological kidney sections stained for PLVAP (magenta) and the nuclear marker 4′,6-diamidino-2-phenylindole (DAPI, cyan). The dashed line borders the glomeruli. Scale bar: 50 µm.

**Figure 2 ijms-24-01094-f002:**
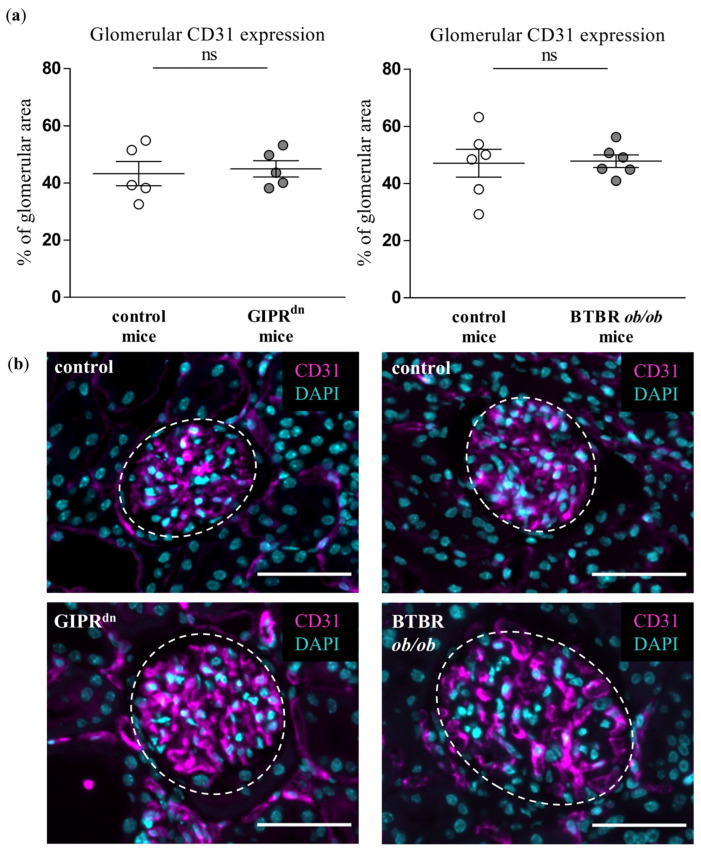
Analysis of CD31 in diabetic kidney disease models. (**a**) Automatic histological evaluation of CD31-positive area in glomeruli of GIPR^dn^ mice (**left**, diabetes mellitus type 1 model) and BTBR *ob/ob* mice (**right**, diabetes mellitus type 2 model) compared to their corresponding healthy control mice. Scatter dot plot with means and SEM, ns: not significant, *n* = 5–6. (**b**) Representative images of histological kidney sections stained for CD31 (magenta) and the nuclear marker 4′,6-diamidino-2-phenylindole (DAPI, cyan). The dashed line borders the glomeruli. Scale bar: 50 µm.

**Figure 3 ijms-24-01094-f003:**
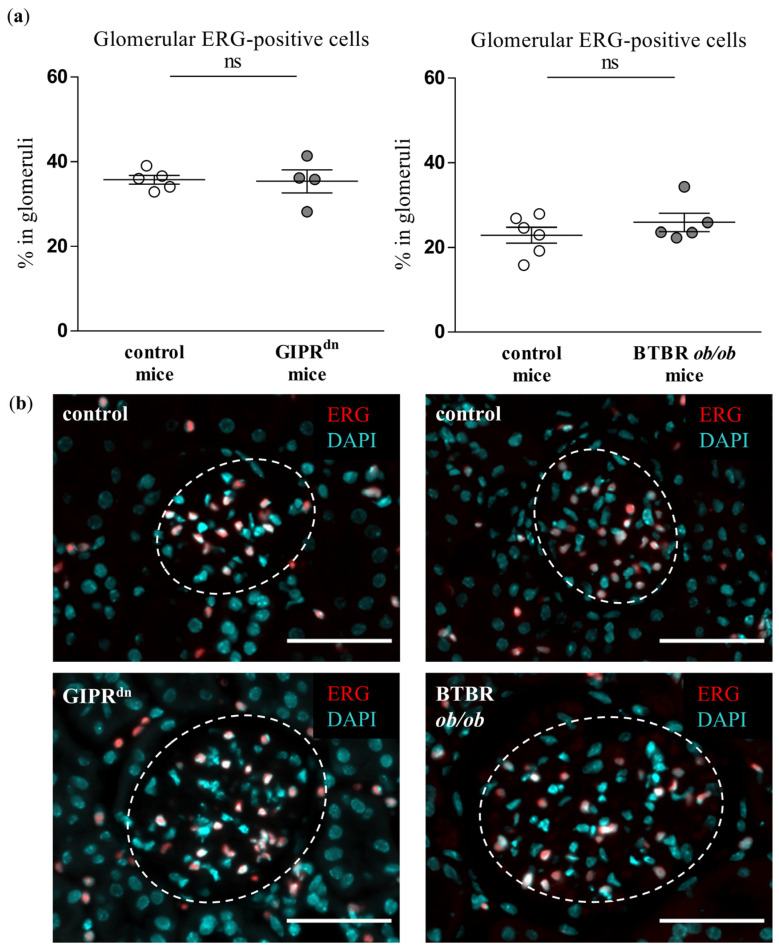
Analysis of the erythroblast transformation-specific related gene (ERG) in diabetic kidney disease models. (**a**) Automatic histological evaluation of ERG-positive cells in glomeruli of GIPR^dn^ mice (**left**, diabetes mellitus type 1 model) and BTBR *ob/ob* mice (**right**, diabetes mellitus type 2 model) compared to their corresponding healthy control mice. Scatter dot plot with means and SEM, ns: not significant, *n* = 5–6. (**b**) Representative images of histological kidney sections stained for ERG (red) and the nuclear marker 4′,6-diamidino-2-phenylindole (DAPI, cyan). The dashed line borders the glomeruli. Scale bar: 50 µm.

**Figure 4 ijms-24-01094-f004:**
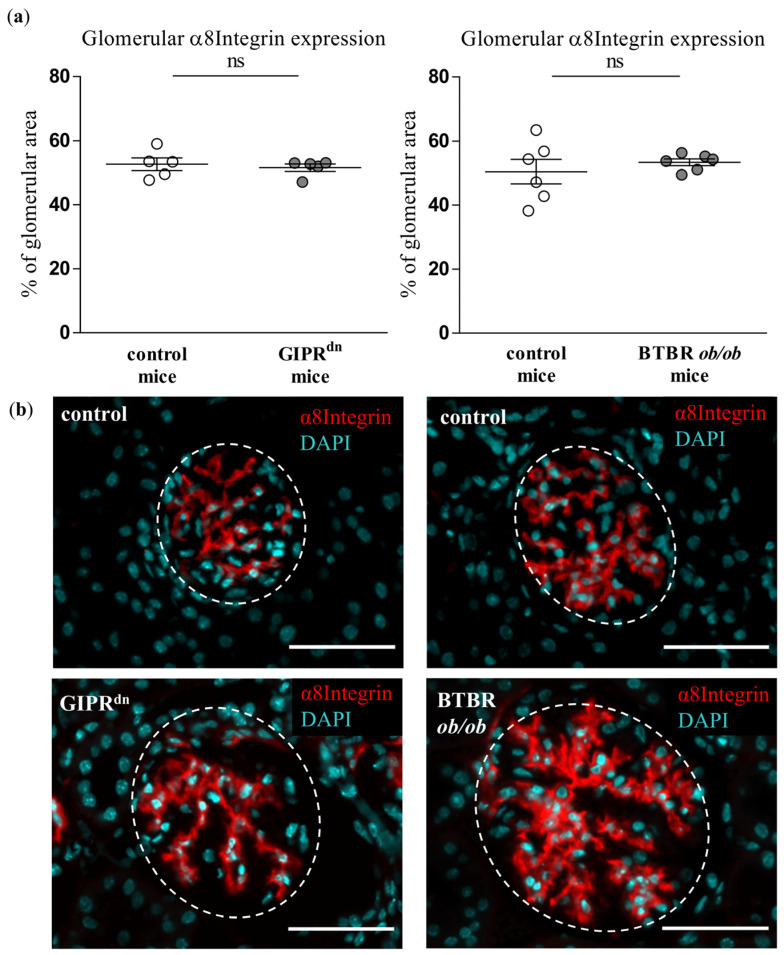
Analysis of α8Integrin in diabetic kidney disease models. (**a**) Automatic histological evaluation of α8Integrin-positive area in glomeruli of GIPR^dn^ mice (**left**, diabetes mellitus type 1 model) and BTBR *ob/ob* mice (**right**, diabetes mellitus type 2 model) compared to their corresponding healthy control mice Scatter dot plot with means and SEM, ns: not significant, *n* = 5–6. (**b**) Representative images of histological kidney sections stained for α8Integrin (red) and the nuclear marker 4′,6-diamidino-2-phenylindole (DAPI, cyan). The dashed line borders the glomeruli. Scale bar: 50 µm.

**Figure 5 ijms-24-01094-f005:**
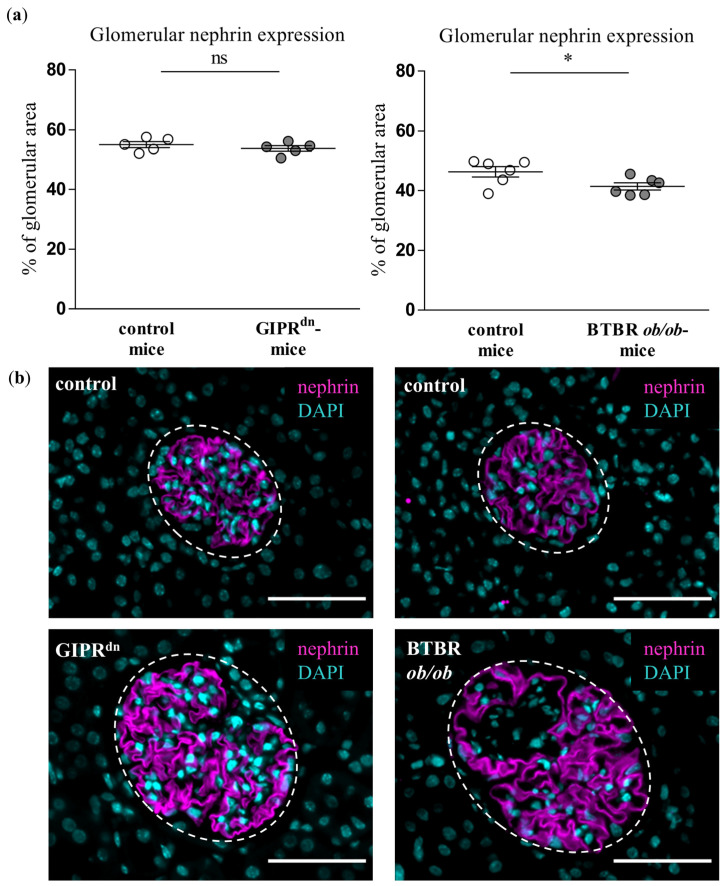
Analysis of nephrin in diabetic kidney disease models. (**a**) Automatic histological evaluation of nephrin-positive area in glomeruli of GIPR^dn^ mice (**left**, diabetes mellitus type 1 model) and BTBR *ob/ob* mice (**right**, diabetes mellitus type 2 model) compared to their corresponding healthy control mice. Scatter dot plot with means and SEM, * *p* < 0.05, ns: not significant, *n* = 5–6. (**b**) Representative images of histological kidney sections stained for nephrin (magenta) and the nuclear marker 4′,6-diamidino-2-phenylindole (DAPI, cyan). The dashed line borders the glomeruli. Scale bar: 50 µm.

**Figure 6 ijms-24-01094-f006:**
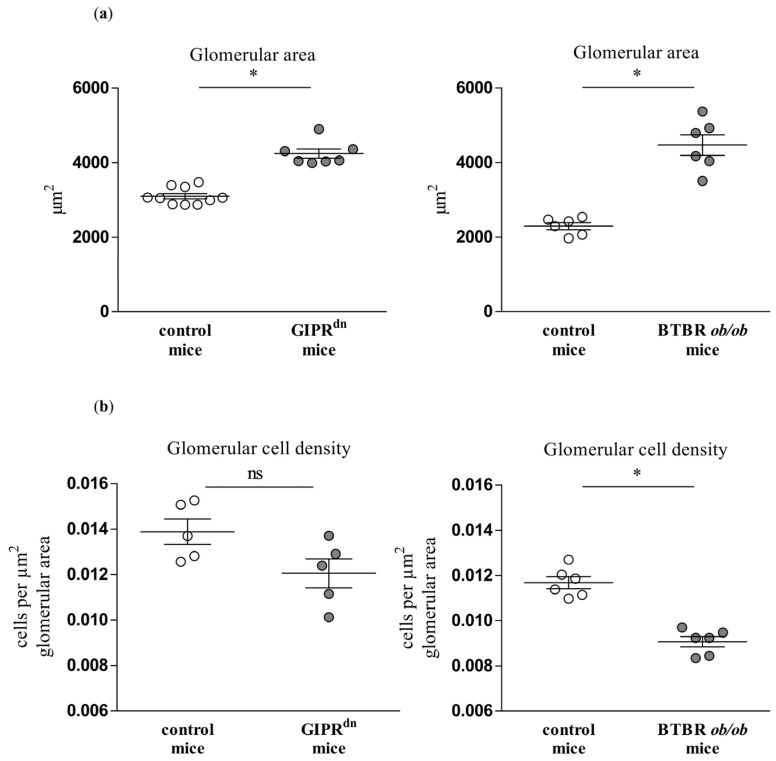
Analysis of glomerular morphology in diabetic kidney disease models. Automatic histological evaluation of glomerular area (**a**) and glomerular cell density (**b**) in kidneys of GIPR^dn^ mice (**left**, diabetes mellitus type 1 model) and BTBR *ob/ob* mice (**right**, diabetes mellitus type 2 model) compared to their corresponding healthy control mice. Scatter dot plot with means and SEM, * *p* < 0.05, ns: not significant, *n* = 5–10.

## Data Availability

The data presented in this study are available on request from the corresponding author.

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
