# Peer review of "PLVAP as an Early Marker of Glomerular Endothelial Damage in Mice with Diabetic Kidney Disease"

_ijms, 2023, doi:10.3390/ijms24021094_

Round 1
Reviewer 1 Report
This study explored the possibility of PLVAP as a marker of glomerular endothelial damage in DKD, which is a comprehensive and rigorous study and worthy of publication. However, this manuscript remains a handful of mistakes. I suggest that there should be a minor revision before it is accepted for publication.
Comment 1: Is there SEM data available to corroborate the results of the automatic histological evaluation? Please upload your experimental data as support.
Comment 2: Is there data on glomerular morphology used to corroborate result 2.4? Please upload your experimental data as support.
Comment 3: Is there data to support the dose selection of STZ? Please upload your experimental data as support.
Comment 4: Is blood released during kidney sample collection? How do you ensure that kidney samples are not affected by blood?
Comment 5: Please check and correct the incorrect punctuation in Figure S3.

Reviewer 2 Report
This study aims to provide an early biomarker for the detection of glomerular endothelial damage in mice models. Comparing to the known markers such as CD31 or erythroblast transformation-specific related gene (ERG), PLVAP did express much earlier in both type 1 and 2 DM models. This is a meaningful finding for the diagnosis of DKD while some suggestions/questions are raised by the reviewer.
1. Abstract: please avoid using too many “we” in scientific writing.
2. Based on S1 Table, some of the primary antibodies used in this study are not specific for the mouse antigens (including PLVAP), the authors need to prove these antibodies react exactly with the target antigens.
3. The authors did not provide details about when the urine and serum samples were collected for the analysis.
4. The authors may need to further interpret the clinical application of PLVAP in the discussion section since it will be more favorable for DM or DKD patients if this marker could be detected within the body fluids.
Round 2
Reviewer 2 Report
The responses from the authors can be accepted.